



**Attributing Regional Trends of Evapotranspiration and Gross**
**Primary Productivity with Remote Sensing: A case study in the**
**North China Plain**
Xingguo Mo[1,2], Xuejuan Chen[1,2], Shi Hu[1], Suxia Liu[1,2], Jun Xia[1]
1. Key Laboratory of Water Cycle & Related Land Surface Processes, Institute of
Geographic Sciences and Natural Resources Research, Chinese Academy of Sciences,
Beijing 100101, China.
2. University of Chinese Academy of Sciences
Correspondence to: Xingguo Mo, E-mail: moxg@igsnrr.ac.cn
**Abstract**
Attributing changes of evapotranspiration (ET) and gross primary productivity (GPP) are
crucial for impacts and adaptation assessment of the agro-ecosystems to climate change.
Simulations with VIP model revealed that annual ET and GPP were slightly increasing from
1981 to 2013 over the North China Plain. The tendencies of both ET and GPP were upward in
spring season, while the trends are weak and downward in summer season. A complete
factor‐separation analysis illustrated that the relative contributions of climatic change, $CO_2$
fertilization and management to ET (GPP) trend were 56 (-32)%, -28 (25)% and 68 (108)%,
respectively. The decline of global radiation resulted from deteriorated aerosol and air
pollution was the principal causes of GPP decline in summer, while air warming intensified
the water cycle and advanced the plant productivity in the spring season. Agronomical
improvements were the principal drivers of crop productivity enhancement.
Key words: Climate change; Contribution; VIP model; Evapotranspiration; Gross primary
productivity
**1. Introduction**
Terrestrial hydrological and carbon cycles are intimately coupled via transpiration and



photosynthesis processes which are regulated by plant leaf stomata. Due to land use/cover
changes, intensified agricultural management and climatic change, terrestrial
eco-hydrological processes have been noticeably shifted at multiple spatiotemporal scales
(Tian et al., 2011; Douville et al., 2013), for example, prevailing irrigation and application of
chemical fertilizers have raised soil moisture, evapotranspiration (ET) and crop productivity,
etc. In some regions the effects of human activities are the same magnitude as, or even exceed
the impacts of global warming on the productions of agro-ecosystems (Haddeland et al.,
2014). In the last decades, global consumptive water use and carbon fixation by terrestrial
ecosystems are demonstrated to be slightly increasing with more efficient water use,
corresponding to changes of climatic factors and fertilization effect of elevated atmospheric
$CO_2$ concentration (Yan et al., 2013; Nayak et al., 2013). Spatiotemporal patterns of water
and carbon fluxes at regional scale are changing under global change ( Zeng et al., 2014;Liu
et al., 2012).
As ET being the major component of water budget in the water limited basins, its long
term tendency has been taken as an indicator for diagnosing the intensification of regional
water cycle. The complementary relationship between actual and potential ET may reveal
some clues of hydrological changes. Observations in the last decades illustrated that potential
evaporation rates ($ET_p$)(represented as pan evaporation) were decreasing in Europe, U.S.,
China, India, Australia (Brutsaert, 2006; Katul et al., 2012), implicating the decline of
available energy and aerodynamics devoted to latent heat flux over the land surface. The
climatic factors dominating $ET_p$ change are usually diverse. For example, over the North
China Plain (NCP) the changes of $ET_p$ were mainly attributed to declines of global radiation
and near surface wind speed (Tang et al., 2011; Song et al., 2010). However, in southern
Turkey a noticeable decline of $ET_p$ was attributed to enhanced air humidity associated with
the expansion of irrigation acreage and more water evaporated into the atmospheric boundary
layer (Ozdogan and Salvucci, 2004). Burn and Hesch (2007) revealed that decreasing wind
speed and raised water vapor deficit were responding to trend of $ET_p$ in Canadian Prairies. At
large scale, precipitation is usually the principal factor determining actual ET changing, such
as Qian et al. (2007) presented that increase of ET in the Mississippi River basin was
following precipitation propensity, while the effects of solar radiation and air temperature



changes were minor.

Terrestrial eco-hydrological processes are driven by climate and modulated by human

activities. Generally climate warming enhances atmospheric evaporative demand, while $CO_2$
fertilization stimulates photosynthesis and inhibits leaf stomatal conductance, leading to more
biomass accumulation and higher water productivity (Field et al., 1995; Buckley and Mott,
2013). Simultaneously, land use change and land management also noticeablly affect the
ecosystem production and hydrological fluxes (Shi et al., 2011). Separating the contributions
of climatic change, $CO_2$ enrichment and human activities to the long term trends of water and
carbon cycles is critical for assessment of ecosystem responses and resilience to
environmental changes. Some researchers have explored the relative contributions of climate
change and vegetation dynamics to changes of global land surface evapotranspiration and
river runoff (Betts et al., 2007; Piao et al., 2007; Alkama et al., 2010; Liu et al., 2012; Chen et
al., 2014; Banger et al., 2015), but the conclusions are inconsistent yet. Climate change
dominated the inter-annual variability of ET, while land use changes and agricultural practices
and techniques exerted more discernable effects on water cycle in long term (Liu et al., 2012).
However, Alkama et al. (2010) and Shi et al. (2011) demonstrated that climate change is the
predominant driver of the changes of global ET in $20^{th}$ century. For    the contributions of
climate change to vegetation productivities at large scale may be explored by ecosystem
models or statistical models. Piao et al. (2015) documented that elevated atmospheric $CO_2$
and nitrogen deposition were the critical contributors to terrestrial greening over China in the
last three decades; Baker et al. (2010) figured out that climate anomalies in springtime were
the most frequent drivers to annual GPP variability in the North America; Nayak et al. (2013)
reported that climate change had a relatively small but significant control (15%) on the trend
of terrestrial net primary production (NPP) over India during 1981to 2005. In the crop
ecosystems, contributions of climate change, cultivar renewal and agronomic management to
change of crop yield have been separated with crop or statistical models (Yu et al, 2012; Song
et al.,2014; Bai et al., 2015; Guo et al., 2014; Wang Z. et al., 2016).    The impacts of climate
change on crop yield may be positive or negative in different regions, depending on the
tendencies of the dominant factors (Ewert et al., 2015).

As one of the granaries in China, North China Plain (NCP) is experiencing challlenges of



agriculture sustainability due to global change and social development. Thereby, it is crucial
to understand the impacts of climate change on the productions of cropping systems. Over the
plain, winter wheat – summer maize double cropping system is prevailing, supported by
irrigation, fertilizer and agronomical techniques. *In situ* measurements, agricultural annals and
regional remotely sensed vegetation index dataset all illustrated that both wheat and maize
productivities have enhanced remarkably during the last three decades (Yuan and Shen, 2013);
correspondingly, the seasonal water consumption and water use efficiency are also slightly
improved (Zhang et al., 2011). The achievement of long term increasing grainproduction is
related to the active adoption of new varieties for stabilizing, extending the length of crop
growth period, as well as agronomical technique advancement (Liu et al., 2010; Sacks and
Kucharik, 2011). Currently, water amount for food production is consisted of 65% of total
water consumption here. Further, along with the gradually augmented domestic and industrial
water requirement, groundwater in some parts of the plain has been over-exploited, and the
environmental water requirement is generally under deficit conditions (e.g., MWR, 2010).
Facing with the rapid deteriorating agricultural environment, some critical issues are still
unclear, such as, what mechanisms drive the evolutions of eco-hydrological processes over
this plain? What are the impacts of climate change on the cropping systems?

In this study, the VIP eco-hydrological dynamic model integrated with NOAA-AVHRR

remotely sensed normalized difference of vegetation index (NDVI) is employed to assess the
spatiotemporal evolutions of ET and vegetation GPP over the NCP during 1981 to 2013. By
numerical experiments with the VIP model and the factor separation method, the
contributions of climate change, fertilization of atmospheric $CO_2$ enrichment and agronomical
practices and technological advancement to crop water consumption and productivity are then
analyzed, and the relevant mechanisms are discussed.

## 2. Method and materials

### 2.1 Study site

The NCP is one of the country's granaries, extending from latitude 32°00′ to 40°24′N

and longitude 112°48′ to 122°45′E (Fig. 1(a, b)). It is located in the eastern part of China with




an area of $33\times10^4$ km$^2$, which is an alluvial plain developed by the intermittent flooding of the
Huang, Huai and Hai Rivers and 72% is cultivated as farmland. The warm temperate climate
varies gradually from sub-humid in the southern to semi - arid in the northern parts. The
annual precipitation ranges from 500 - 1000 mm, occurring irregularly among seasons and
more than 70% falls in summer. Soil moisture deficit happens widely during the the spring
and early summer period. Besides soybean/millet/sorghum/cotton, the double cropping
system of winter wheat - summer maize is prevailing in the plain, where wheat and maize are
the most common harvest crops in summer and autumn seasons, respectively. Due to
insufficient precipitation, the spring crops (such as wheat) usually need supplemental
irrigation to form favorable production.

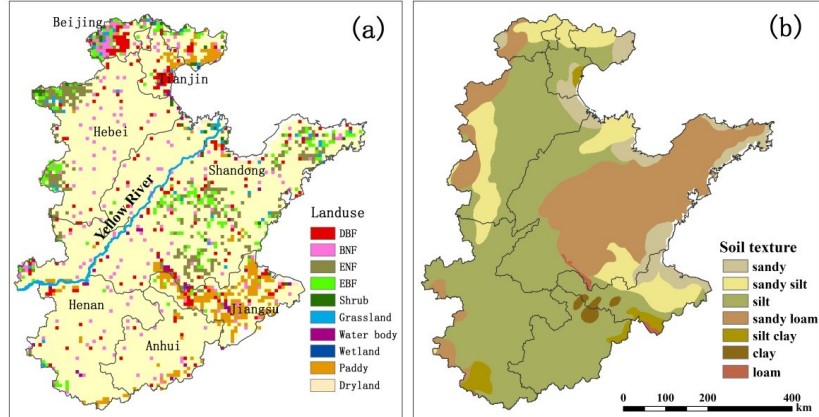



Fig. 1 Land use/cover (a) and soil texture (b) of the North China Plain (NCP) (DBF:
Deciduous Broadleaf Forest; BNF: Broadleaf and Needle leaf Mixed Forest; ENF: Evergreen
Needle leaf Forest; Evergreen Broadleaf Forest)

**2.2 The VIP eco-hydrological Model**

The physically process-based VIP (Vegetation Interface Processes) eco-hydrological model
is designed to simulate the exchanges of energy, water and carbon between terrestrial
ecosystem and atmosphere (Mo et al., 2014). In the model, ET is termed as the summation of
canopy transpiration, evaporation from canopy intercept and soil surface, computed
separately with the Penman-Monteith Equation. Transpiration and photosynthesis processes





are coupled through the Ball-Berry relationship between leaf stomatal conductance and net
assimilation rate. On carbon cycle aspect, leaf carbon fixations on sunlit and shaded leaves
groups are predicted with the biochemical schemes for C3 (Farquhar et al., 1980) and C4
plants (Collatz et al., 1992). In the radiation budget scheme, shortwave radiation transfer in
canopy distinguishes the leaf spectral properties of visible and near infrared radiation, as well
as fraction of direct beam and diffusive irradiance in global radiation. Precipitation
throughfall - runoff generation over the land surface is calculated with a curve - number (CN)
type equation at daily scale, using the daily net precipitation and the moisture deficit of
upper-soil layer in this study. Simulation of soil water movement in root zone is carried out
with a discrete Richards Equation in three layers. The crop and natural vegetation growth
modules are also embedded in the model to simulate the biomass mass accumulation and
carbon cycle.
**2.3 Data**
The VIP model input data include land use/cover, soil physical properties, and atmospheric
forcing variables. GIMMS AVHRR 15-day normalized difference of vegetation index time
series (NDVI3g) from 1981 to 2013 is used to retrieve the vegetation leaf area index and
other land surface characteristics (https://nex.nasa.gov/nex/projects/1349/wiki/general_data_
description_and_access/) (Pinzon and Tucker, 2014). The land use classification is originated
from both Landsat TM images (www.geodata.ac.cn) and MODIS remote sensing products, in
which the farmland is classified as rice paddy and dryland. Soil textural data are at a scale of
1:1,000,000 represented as fractions of sand, silt and clay, by which the parameters of soil
porosity ($\theta_{sat}$) and saturated hydraulic conductivity ($K_{ws}$, mm s$^{-1}$) are estimated as Bonan
(1996). Daily climate variables (air temperature, water vapor pressure, wind speed, sunshine
duration and precipitation) recorded at 87 climatic stations (http://cdc.cma.gov.cn/) in and
around the study area are available for generating the spatial atmospheric forces. The NDVI
data are error-checked and the erroneous data are replaced by interpolation with the preceding
and subsequent values according to the time series by the Savitzky–Golay (SG) filter
(Savitzky and Golay, 1964), and then the daily values are derived with the Lagrange
polynomial. Vegetation leaf area index (LAI) is retrieved from NDVI with empirical



relationships for different plant function types.
The data used for model validation are field flux measurements with eddy covariance
technique at Yucheng (116°38′E, 36°57′N), Daxing (116°25′E, 39°37′N), Miyun (117°19′E,
40°38′N) and Guantao (115°8′E, 36°31′N) sites over the plain. The cropping systems at the
Yucheng, Daxing and Guantao sites are all rotations of winter wheat – summer maize, while
land cover is dwarf shrub at the Miyun site. The eddy covariance data are processed with
general procedures (Liu and Xu, 2013). GPP data are available only at Yucheng site. In
addition to the eddy fluxes, grain yield records of wheat and maize in county statistics are also
used to verify the GPP predictions at regional scale.

## 2.4 Model implementation and experimental design

### 2.4.1 Simulation setup

The model simulations were conducted at 8 - km spatial resolution and half - hour time step.
The cropland is classified into wheat and maize or rice double cropping systems. Atmospheric
driving forces are interpolated from daily meteorological variables recorded at the climatic
stations to grid cells with a gradient inverse distance square method (GIDS), which accounts
for the effects of elevation, latitude and longitude (Nalder and Wein, 1998). Estimated with
sunshine duration in a linear relationship, the global radiation is subdivided into direct visible
and near infrared parts, as well as direct beam and diffusive components with Weiss and
Norman (1985). The daily air temperature is extended to hourly values with a sinusoid
function based on the daily maximum and minimum temperatures (Cambell and Norman,
1998). During the winther wheat growing period, irrigation water is supplied when water
storage in the root zone is below 60% of the field capacity. Summer maize is set to be
irrigated not more than one time in its growth period. The simulation is conducted with
prescribed daily LAI series retrieved from remotely sensed NDVI series for eco-hydrological
prediction from 1980 to 2013, in which the first year is taken as warming up.

### 2.4.2 Separation of the contributions of climate change and management effects

By using a general function, $f$, the scalar fluxes (water vapor and carbon) between land





surface and the atmosphere are determined by climate factors (M), atmospheric $CO_2$
fertilization (C) and agronomical management and technological advancement (In this study
we assume   the long term trend of leaf area index (LAI) may represent the effects of human
activies on ecosystems of crop and natural ecosystems. Human activities to the
agro-ecosystem include renewals of cultivars, irrigation facility improvement, fertilizer use
application, soil quality amelioration, etc.), namely,
$$f = F(M, C, LAI, \ldots) \tag{1}$$

The changes of $f$ contributed by a single factor (expressed as $f_i$) and its interaction with
another factor (expressed as $f_{ij}$) can be decomposed by the Taylor expansion as,
$$f_i = \frac{\partial F}{\partial x_i} \Delta x_i + \frac{1}{2!} \frac{\partial^2 F}{\partial x_i^2} \Delta x_i^2 + \cdots + \frac{1}{n!} \frac{\partial^n F}{\partial x_i^n} \Delta x_i^n \tag{2}$$

$$f_{ij} = f_i + f_j + \frac{\partial^2 F}{\partial x_i \partial x_j} \Delta x_i \Delta x_j + \cdots \tag{3}$$

where $x_i$ and $x_j$  ($i \neq j$) represent $M$, $C$ and $LAI$, respectively. The factor separation methodology
from Stein and Alpert (1993) and Alkama et al. (2010) is used to category the contributions of
climate change, $CO_2$ fertilization and $LAI$, and their interactions to long term trends of ET and
GPP. Similar to Alkama et al. (2010), the total effect, $f_{123}$, is expressed as,
$$f_{123} = f_1 + f_2 + f_3 + f^{12} + f^{13} + f^{23} + f^{123} \tag{4}$$

With
$$f^{12} = f_{12} - f_1 - f_2 \tag{5}$$

$$f^{13} = f_{13} - f_1 - f_3 \tag{6}$$

$$f^{23} = f_{23} - f_2 - f_3 \tag{7}$$

Where $f_1$, $f_2$ and $f_3$ are the direct contributions of climate change, atmospheric $CO_2$ enrichment
fertilization and agronomical management, respectively; $f^{12}$ is the contribution of interactions
of climate change and $CO_2$ enrichment; $f^{13}$ is the contribution of interactions between climate
change and management; $f^{23}$ is the contribution of interactions between $CO_2$ fertilization and
agronomical management; $f^{123}$ is the contribution of interactions between climate change, $CO_2$
fertilization and agronomical management.
Seven numerical experiments designed to fully distinguish the contributions of climate
change, $CO_2$ fertilization and agronomical management are conducted by the VIP model over
the NCP from 1981 to 2013. The experiments are as following:





(1) $f_{123}$ ("all factors"): Current climate, $CO_2$ and LAI spatiotemporal pattern;
(2) $f_1$ ("climate change effect"): Current climate, but atmospheric $CO_2$ concentration is fixed
at year 1981, and LAI pattern is set as the multi-year average;
(3) $f_2$ ("$CO_2$ fertilization effects"): Climate and LAI fixed at a specific year, but current $CO_2$
concentration;
(4) $f_3$ ("management effect"): Climate and $CO_2$ concentration are fixed a specific year, but
current LAI pattern is used;
(5) $f_{12}$ ("climate change and $CO_2$ fertilization effects"): LAI pattern is fixed, but current
climate and $CO_2$ concentration are used;
(6) $f_{13}$ ("climate change and management effects"): $CO_2$ concentration is fixed, but current
climate and LAI pattern are used;
(7) $f_{23}$ ("$CO_2$ fertilization and management effects"): Climate is fixed at 1981, but current
$CO_2$ and LAI are used.
The trends of annual ET and GPP in the above experiments are calculated. According to
Eq.(4) to Eq.(7), the contributions of climate change, $CO_2$ fertilization and management to ET
and GPP long term trends are separated.
**3. Result analysis**
**3.1 Model Verification**
**3.1.1 Validated with eddy covariance measurements**
The VIP model is used to simulate the hydrological, energy partitioning and crop growth
processes at the four sites of eddy flux measurement. Here, eddy covariance measurements of
daily ET and GPP are employed to verify the model predictions (ET is available in all the
sites, but GPP is only available at one site). The land surface characteristics are relatively
homogeneous surround the measuring sites, ensuring the footprint for measured fluxes. The
meteorological information measured at each site is used to drive the VIP model. It is shown
that the agreements are quite satisfactory for both ET and GPP (Fig.2 and Fig.3). Totally, there





are 9-year daily ET data and 3-year daily GPP data for comparison with the model
simulations. The coefficients of determination ($R^2$) are above 0.76 for all the sites. At annual
scale, the absolute relative biases of predicted ET are ranged from 1.5 to 12.6% in the 9-year
dataset, and biases of GPP are from 2.0 to 8.8% in 3-year data. Therefore, the model
performance is quite well and reliable for vegetation/crop productivity and water consumption
predictions. The biases may be stemmed from both meausrements and model uncertainty. Mo
et al. (2012) showed that the canopy leaf area index (LAI) and photosynthetic capacity
(carboxylation rate for C3 crops and photon quantum use efficiency for C4 crops) were the
most sensitive parameters to the model efficiency. Here, taking Yucheng site as an example,
annual ET and GPP may increase 1.6% (2.6%) and 3.0% (15.9%) respectively as LAI
(photosynthesis capacity) is increased by 20%.

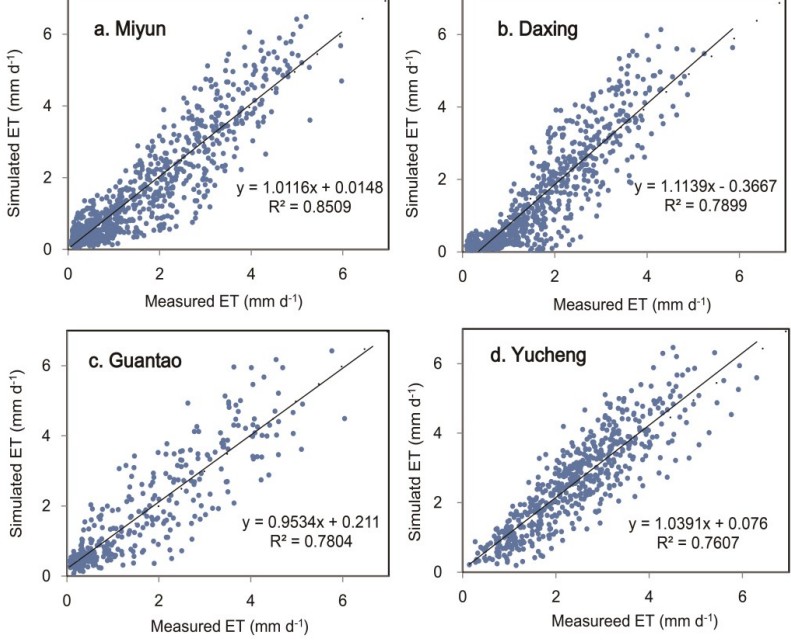

Fig.2 Comparison of the simulated daily ET and GPP with the eddy covariance
measurements




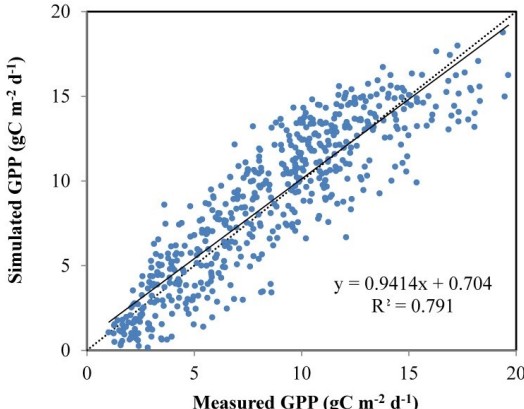


Fig.3 Comparison of the simulated GPP with eddy covariance measurements at the Yucheng
site
**3.1.2 Validated with the statistical yield records**

The simulated GPP is also validated with the statistic staple crop grain yields at county

level. The yield per hectare is converted to equivalent GPP per square meter. As shown in
Fig.4 (years of 2000 and 2005 are used), the agreement is satisfactory with the coefficient of
determination ($R^2$) of 0.43 and 0.51 ($p<0.001$), respectively. There are remarkable spatial
variations of crop yields resulted from diverse conditions of climate, soil and management. In
the simulation, it is found that the spatiotemporal evolution of greenness is the dominant
factor of yield patterns. Greenness represented by the vegetation index is an appropriate
indicator of crop productivity under environmental stresses (Hu et al., 2014). In the areas with
high vegetation index and favorable irrigation facilities, the yield losses may be caused by
heat waves or pest infections in the maturity stage.





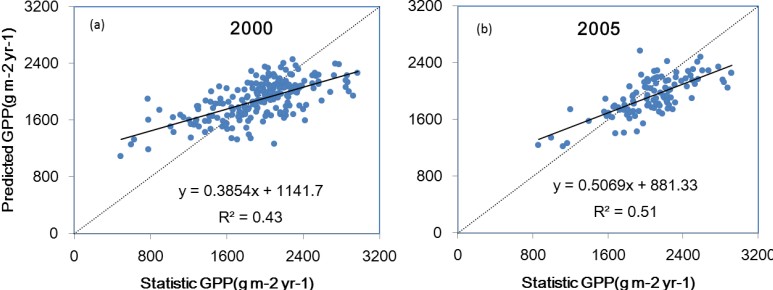


Fig.4 Comparison of the predicted GPP and statistic yield derived GPP at county scale in

2000 and 2005.

## 3.2 Trends of climate, crop productivity and ET

Changes of climate variables and agro-ecosystem management are the dominant driving

forces for evolution of regional eco-hydrological processes. Intra-seasonal variations of
climatic variables may exert different impacts on the crop water consumption and carbon
assimilation. In the last three decades, air temperature is rising, but sunshine duration and
wind speed are decreasing significantly over the plain, associated with global climate change,
aerosol and air pollutions. Soil amelioration, genetic improvement, irrigation facility
constructions and application of chemical synthesis fertilizer are considered to be the
principal factors that have propelled productictiy close to the attainable level(Yu et al., 2012;
Lobell and Burke, 2010).

### 3.2.1 Changes of climatic variables

Grid averages of the climatic variables were interpolated with GIDS (Gradient Inverse

Distance Square) method over the North China Plain from 1980 to 2013. Nevertheless
inhomogeneous distributions of the climatic variables, the spatially averaged trends were
clear (Table 1). At annual scale, global radiation, air temperature (especially minimum
temperature) and wind speed were significantly changing ($p<0.01$). At monthly scale,
radiation was declining in all the months except March, but only trends in June to September
were significant ($p<0.01$); Significant increasing of air temperature is occurred in spring
(February and March) and early summer (May to July); Wind speed was decreasing



significantly ($p<0.01$) in all the months except August. However, no significant trends were
detected for both precipitation and water vapor pressure throughout each month. As a
consequence, water vapor pressure deficit was exaggerated along with the rising of air
temperature, which was expected to intensify the atmospheric water vapor demand and offset
the negative effects of declining radiation and wind speed on potential evaporation ). These
changes in climatic variables have exerted remarkable impacts on the crop phenological
stages, water consumption and productivity during the last three decades over the North
China Plain (Liu et al., 2010).

Table 1. Inter-annual trends of monthly sunshine duration, precipitation, air temperature, relative
humidity and wind speed (Significant levels: * for $p<0.05$; **for $p< 0.01$).

| | Jan | Feb | Mar | Apr | May | Jun | Jul | Aug | Sept | Oct | Nov | Dec |
|---|---|---|---|---|---|---|---|---|---|---|---|---|
| Sun(h/yr) | -0.026 | -0.036* | 0.015 | -0.012 | -0.015 | -0.039** | -0.040** | -0.052** | -0.060** | -0.018 | -0.017 | -0.020 |
| P(mm/yr) | -0.129 | 0.212 | -0.279 | 0.354 | 0.049 | -0.472 | 0.608 | 0.456 | 0.694 | -0.706 | 0.223 | 0.084 |
| T(°C/yr) | 0.022 | 0.058* | 0.078** | 0.030 | 0.042** | 0.029* | 0.030* | 0.017 | 0.020 | 0.044* | 0.025 | 0.018 |
| rh(%/yr) | -0.065 | 0.115 | -0.260* | -0.079 | -0.139 | -0.074 | -0.080 | -0.074 | 0.057 | -0.125 | -0.094 | -0.091 |
| U(m/s/yr) | -0.016** | -0.012** | -0.012** | -0.020** | -0.021** | -0.019** | -0.015** | -0.006 | -0.014** | -0.017** | -0.015** | -0.012** |


### 3.2.2 Changes of Greenness and GPP

Here remotely sensed NDVI is expressed as vegetation greenness. Averaged over the
growth period (from March to October), vegetation greenness was significantly increasing
from 1980s with a trend of 0.64/yr ($p<0.001$) and coefficient of variation (CV) of 2.4%
(Fig.5(a, b)). It was noted that the maximum inter-annual variation of greenness was 5.4%
between two consecutive dry and wet years (1989 and 1990) with 280 mm difference of
annual precipitation. The distinguish differences of greenness occurred in June to December.
At annual scale greenness was weakly related with precipitation; however, in growing season,
greenness was noticeably correlated with monthly precipitation in April, May, June,
September and November (r =0.29 ~ 0.53, $p<0.1$). The reason is that the monthly rainfall is





generally lower than the atmospheric evaporative demand in spring season, and the water
deficit to transpiration generally stresses the plant growth. Unlike precipitation, greenness
anomalies are positively correlated with the detrended air temperature ($r^2$=0.16, $p$<0.05),
implicating that recent climate warming has stimulated vegetation growth through extending
the growing stage and through pushing photosynthesis in water no-limited regionss (Mao et
al., 2012; Nemani et al., 2003).
Regional greenness trends showed remarkably diverse (Fig.6(a, b)). Except climate change,
human activities also exerted critical impacts on the land greenness variations. In the low
plain of Hebei province, saline - alkali land amelioration and irrigation facilities improvement
have contributed greatly to the greenness enhancement in the 1980s to 1990s. In addition,
atmospheric nitrogen deposition was also regarded as a positive driver for the land greening,
since the nitrogen deposition has averagely increased by 25% from 1990s to 2000s in North
China (Jia et al., 2014; Piao et al., 2015). Spatially, greenness over 91.3% of the NCP was
increasing, in which the most distinctive grids distributed in the southern parts and the belt
along the yellow river channel, where water supply was usually sufficient. In the northern part,
tendencies of greenness in a number of grids were decreasing significantly at $p$=0.1 level,
which were resulted from less irrigation supply to farmland in springtime and rapid expansion
of built-up occupations around cities and towns, such as Beijing and Tianjin metropolitans.

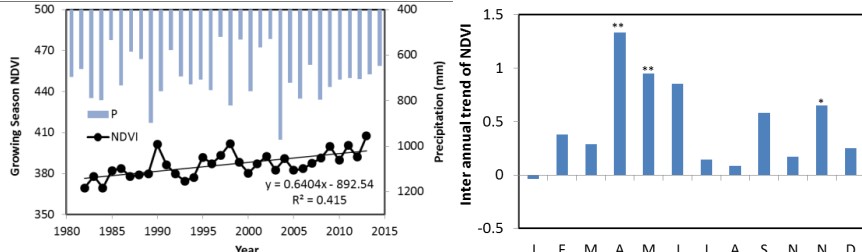

Fig.5 Spatial average trends of growing season NDVI at annual (a) and monthly scales (b)
(Significant levels: * is $p$<0.05; **is $p$< 0.01).





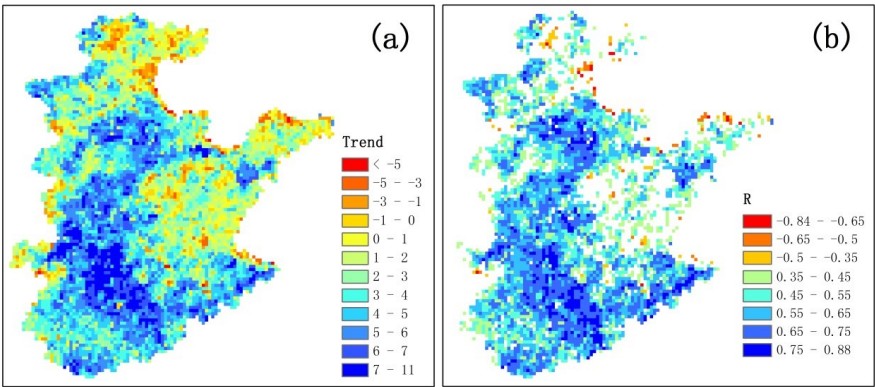


Fig.6 (a) Spatial distributions of NDVI trend in growing season and (b) the Pearson
coefficients of NDVI trend above $p < 0.05$ significant level

The spatially averaged GPP was $1913 \pm 584$ gC m$^{-2}$yr$^{-1}$ with CV of 6.8% predicted by the
VIP model from 1981 to 2013, showing great spatial variability (Fig.7). Low crop
productivity was resulted from fields with saline-alkali soil in the low lands nearby the coast
of Bohai Sea, where almost no favorable water was available for irrigation purpose in
springtime. Averagely, the increasing trend of GPP was significant with a slope of 8.2 gC m$^{-2}$
yr$^{-2}$ ($r=0.60$, $p<0.01$). It was noticed that the average annual GPP was increasing steadily from
1980s to 2000s, compounded by decadal variations of the climate and elevated atmospheric
$CO_2$, as well as the improvement of agricultural practices and techniques. Trends of annual
GPP were positive over 87.9% of the study region. As shown in Fig.7, the obvious increasing
trends were located in the mid and southern areas, while most of the decreasing trends
occurred in the eastern and northern parts, where water for irrigation was considerably
reduced in spring season because of competing demand of the domestic and industrial water
uses.
At monthly scale, GPP was increasing in all the months except July, August and
September (Fig.8). The positive trends were contributed principally by the summer harvest
crops (wheat as the main crop), while the negative trends were mainly contributed by the
autumn harvest crops (maize as the major crop). Regressive analysis showed that the
downward trends of GPP in summer season were resulted from the significant declines of
monthly sunshine duration and radiation ($r=0.38$ to $0.57$ from June to August, $p<0.05$).





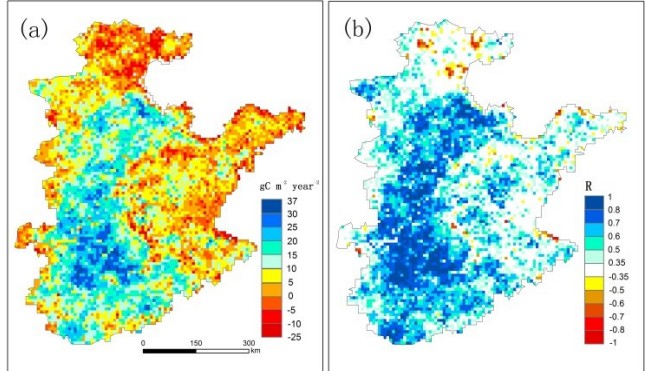


Fig.7 (a) Spatial distribution of GPP trends and (b) Pearson coefficients of trend at p<0.05

significant level.



Fig.8 Monthly trend of GPP from 1981 to 2013 (Significant levels: * is *p*<0.05; **is *p*< 0.01).


### 3.2.3 Evapotranspiration (ET)

Water loss from the vegetation surface as ET is directly regulated by atmospheric vapor

demand and leaf stoma physiologically functioning (Buckley & Mott, 2013). Inter-annual

variation of ET is controlled by climate variability/change and agronomical managements.

Generally potential ET ($ET_p$) is used to represent the available energy for water vaporization

on land surface. As shown in Fig.8a, $ET_p$ was slightly decreasing over the plain during the last

three decades, resulted from offsetting among the effects of reduced global radiation,

declining wind speed and increasing water vapor deficit (Song et al., 2009); simultaneously,




actual ET was predicted to be slightly increasing ($p<0.05$), consistent with the enhancement
of greenness. It was noticed that the evolutions of potential and actual ET coincided with the
hypothesis of complementary relationship. At monthly scale, ET was significantly increasing
from February to April, but it was decreasing in August (Fig. 8b). This implicates that climate
warming may be beneficial to spring crops by waking wheat recovering early from dormancy,
whereas decline of net radiation ($R_n$) (especially in August, significant level $p<0.001$) may
lead to the downward tendency of ET rates in summer.
Over the whole plain, spatially averaged actual ET and transpiration were 627±162 mm
$yr^{-1}$ (about 92% of annual precipitation) and 416±129 mm $yr^{-1}$ (about 67% of ET),
respectively. The trend of annual ET ($p<0.1$)with CV (coefficient of variation) of 0.05 was
0.88 mm $yr^{-2}$ from 1981 to 2013, which was less significant than that of NDVI ($p<0.01$).
Decadal ET amounts in 1980s, 1990s and 2000s were 610, 626 and 640 mm, respectively,
corresponding to the slightly rising trend of precipitation. It is found that GPP increased with
higher significant level thant that of ET, implicating the enhancement of water productivity in
the plain. Spatially, the trends of ET were positive over 86.0% of the study region,
distinguishing in the mid and southern parts, while negative trends were mostly occurred in
the northern part (Fig.9(a, b)), which was consistent with the pattern of GPP tendencies. By
using a water balance model, Zeng et al. (2014) also presented an increasing trend of ET over
the North China Plain from 1982 to 2009.

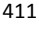
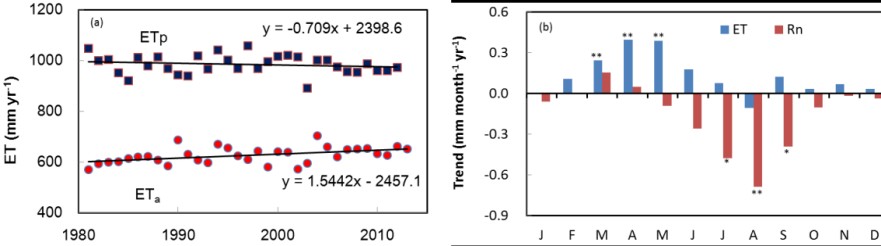



Fig.9 Inter-annual trends of potential and actual annual ET (a) and trends of monthly ET
and net radiation ($R_n$) (b) from 1981 to 2013 (Significant levels: * is $p<0.05$; **is $p<0.01$).





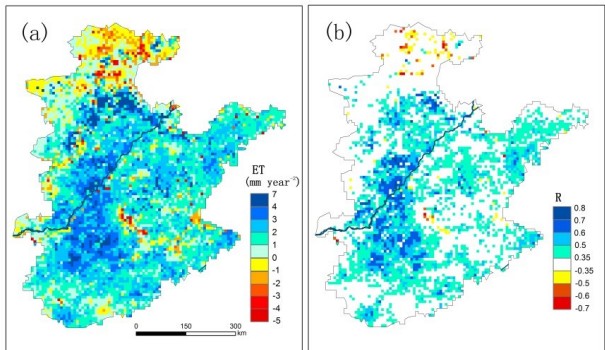


Fig.10 (a) Spatial distributions of ET trends, and (b) their temporal Pearson coefficients

above *p<0.05* significant level from 1981 to 2013.

### 3.3 Contributions of climate change, atmospheric CO$_2$ fertilization and agronomical management to changes of ET and GPP

#### 3.3.1 Spatial patterns of the contributions

The contributions from climate change, atmosphere CO$_2$ enrichment fertilization and
agronomical management illustrated considerably spatial heterogeneity for both ET and GPP
(Fig.11). Over the whole plain, climate change was exerting positive impact on water vapor
exchange from land surface to the atmosphere ($f_1$), especially in the eastern hilly part where
precipitation was increasing slightly. As general knowledge, air CO$_2$ enrichment stimulates
the crop leaf stomatal closing and then reduces transpiration, but its fertilization effect
enhances  photosynthetic rate and water use efficiency (Buckley and Mott, 2013).
Descriptions of the separated effects were presented as following:
The climate change has intensified ET rate almost in the whole area, resulting in 0 to 4 mm
increment per year. The effect of climate change was much stronger in the mid to eastern
zones with high crop productivities, contributed mainly by air temperature increasing. The
contribution of CO$_2$ enrichment on ET is negative in most areas, ranging from 0 to -1 mm per
year. The attributions of agronomical practices and technological advancement represented by
LAI increase are somewhat complex, namely remarkable increase ranged from 0 to 6 mm per





year in the mid-western area where irrigation facilities and soil conditions have been
ameliorated greatly in the recent decades through land consolidation, de-salinization. Renewal
of cultivars and improved agronomical practices also contributed to the ET intensifying
(Zhang et al., 2011). On the contrary, the grids with expansions of built-ups contributed to
negative trends of ET, relating to urbanization and land use changes.

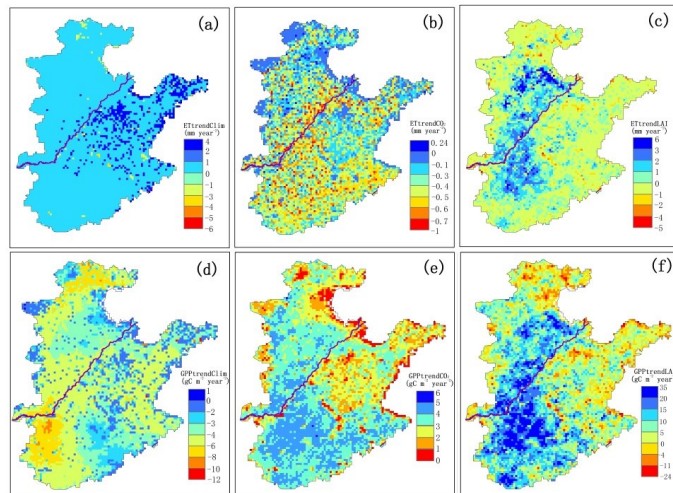



Fig.11 Contributions of climate changes, $CO_2$ enrichment and management on ET (a, b, c) and
GPP (d, e, f) respectively (a and d are for climate; c and e for $CO_2$; c and f are for
management)

Annually, contribution of climate change to GPP is negative, ranged from 0 to -12 gC m$^{-2}$
per year. Lower rates were occurred in the southwestern and northern parts. Air warming and
heat waves, declines of precipitation and global radiation are the main causes of crop
production reduction (Lobell et al., 2011; Guo et al., 2014). In addition, the spatial variability
of climate change effects are associated with the relevant land use/cover and cropping
systems. In the hilly areas (western and mid) and eastern coast areas, negative effects were
slight, where air warming and air pollution were relatively weak. $CO_2$ enrichment effects
were positive over the whole plain, ranged from 0 to 6 gC m$^{-2}$ per year. It was noticed that the
higher effects were associated with higher cropland with favourable irrigation and high
productivity, while the lower rate was related with low productivity croplands and natural
vetetation communities. Similarly, the effects of human activities on ET were positive in the





mid to western areas, ranging from 0 to 35 gC m$^{-2}$ per year, associated with croplands of high
productivity. The negative effects were mainly occurred in the eastern and northern parts
where there is remarkable expansion of urban and dwelling built-ups   in the study period.

### 3.3.2 Regional averaged contributions

On the aspect of regional average, some characteristics of the contributions to water and
carbon assimilation are revealed. As shown in Fig.12(a), the contributions of climatic variable
change ($f_1$), elevated atmospheric $CO_2$ concentration ($f_2$) and agronomical management
(represented by leaf area index (LAI) increment) ($f_3$) and their interactions to the long term
trend of ET were positive, while the contribution of elevated atmospheric $CO_2$ is negative in
the last three decades. It was shown that the contribution of climate change was less than that
of agronomical improvement. The relative direct contributions of climatic change, $CO_2$
fertilization     and     agronomical     management     and     technologic     advancement     to
evapotranspiration long term trend are 56, -28 and 68%, respectively. Compared with the
contributions of direct effects, the relative contributions by their interactions were low (the
cumulative effect of $f^{12}$, $f^{13}$, $f^{23}$ and $f^{123}$ was only about 4%). Although the global radiation
reaching ground was diminished by higher aerosol concentration and deteriorated pollution in
the atmosphere (Che et al., 2005), its negative effect on terrestrial ET was offset by the
positive effects of air warming and higher vapor pressure deficit (VPD) on ET at annual scale.
Reduction of transpiration by enriched atmospheric $CO_2$ caused by closure of plant leaf
stomata at high $CO_2$ concentration for both C3 and C4 plants may mediated the   extra water
demand by air warming. The dominant contribution was from the renewal of cultivars and
improvement of agricultural techniques and management. In the study period, agronomical
management has greatly improved, including the establishment of irrigation facility, prevalent
uses of chemical fertilizers and pesticides. For example, irrigated area in the northern part
(mainly Hebei Plain) has increased by 2.5 times, and chemical synthetic fertilizer input has
increased about four times, consequently, crop grain production has enhanced about two times
from 1980s to 2000s(Xu et al., 2005). Climate change and management improvement
(Irrigation practice, synthetic fertilizers supply and new cultivars adoption) are the main



contributors of ET intensifying over the plain.

As shown in Fig.12(b), the enriched atmospheric $CO_2$ fertilization and agronomical

management improvement presented a positive contribution to GPP trend during the study
period. It was somewhat out of expectation that the contribution of climate change to GPP
was negative at annual scale. The relative contributions of climate change, $CO_2$ fertilization
and management to the vegetation GPP enhancement were -32, 25 and 103%, respectively,
which demonstrated that the improvement of agricultural management was the dominant
driver to GPP increasing in recent decades. The positive effects on GPP were associated with
human activities and natural factors, such as input of synthetic fertilizers and atmospheric
nitrogen deposition, irrigation and other agronomical technology improvement, as well as
fertilization of enriched atmospheric $CO_2$. The negative contribution by climate change was
mainly happened in summertime (Fig.8). Since there was less benefit of $CO_2$ enrichment to
summer maize (C4 type), the reduced maize productivity due to global radiation decline was
not fully offset. Some researches, such as Piao et al. (2015) also reported that climate change
was exerting negative impact on the vegetation greening trend in the northern part of NCP
(including Hebei, Beijing, Tianjin Districts); Liu et al. (2010) attributed the reduction of crop
productivity over the NCP to shortening of vegetative growth length under climate warming.
As shown in Fig.4b, it was illustrated by NDVI time series that greenness in summer season
was quite stable; however it is significantly increasing in spring and autumn seasons,
indicating that climate warming was beneficial for crop growing in the cool seasons. In
addition, carbon assimilated by summer crops was larger than that in spring. Thereby, the sign
of GPP annual trend was determined by the trend in summer season. As air temperature
increasing was not so detrimental to maize growth in summer season yet, the decline of
downward shortwave radiation was considered to be responsible for GPP decline.





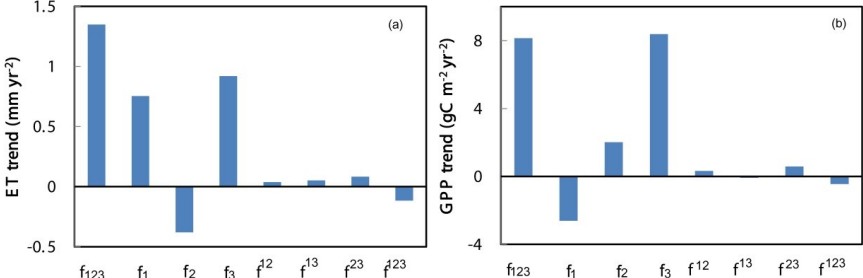



Fig.12(a) Contributions of climate change, atmospheric $CO_2$ enrichment and agronomical
management to ET and (b) contributions to GPP trends

### 3.3.3 Effects of climatic variables on monthly ET and GPP trends

To attribute the responses of cropping systems to the trends of single climatic variables,
the VIP model is used to diagnose the effects of climate change on ET and GPP at Beijing
meteorological observation site. The contributions of a single variable to the trends of ET and
GPP are expressed by their differences simulated with the current and de-trended variables
respectively. Here only the climatic variables of radiation, air temperature and wind speed are
linearly de-trended at monthly scale, since no significant trends of precipitation and humidity
are detected. As shown in Table 2, while the global radiation was de-trended, the negative
correlation coefficient of monthly GPP with time was reversed from negative to positive in
springtime, and from significant (r <-0.6, p<0.01) to insignificant levels (r<-0.15, p>0.01) in
July and August. It was affirmed that the decline of global radiation was the dominant factor
for reduction of crop GPP in summer period (June to August), but in autumn season the
changes of radiation, temperature and wind speed were all responsible for GPP changes. From
Table 2, it could be deduced that the effects of temperature rising on crop productivity was
positive and significant in spring (March and April) and autumn (September and October),
whereas its effect was weak in summer (May to August). It was noticed that the effect of
radiation change was quite weak in March, when no significant trend of shortwave radiation
was detected. In spring season sunshine durations were increasing from 1980s to 2010s
(Wang and Yang,2014). In June, except global radiation, changes oftemperature and





precitpitationhave contributed to GPP increasing. Additionally, fertilizing effect of enriched
$CO_2$ on C3 crop is a critical driver to GPP increasing. However, since the maize as C4 crop
does not benefit much from atmospheric $CO_2$ enrichment, new cultivars with higher light use
efficiency should be adopted to sustain the maize productivity under declined global radiation
condition resulted from exacerbating aerosol concentration and air pollutions.

Comparatively, the effect of climatic change on ET was less significant than that of

vegetation GPP. The model simulations showed that ET enhanced by air temperature rising
was mainly occurring in August to October, while the effect of solar radiation decreasing was
detected from June to September in the maize growing period.

Table 2 Monthly Pearson correlation coefficients of GPP trends (r_ALL: all variable are not
de-trended;r_R: radiation is de-trended;r_T: air temperature is de-trended; r_R-T-U:
radiation, temperature (T) and wind speed (U) are all de-trended).

|  | Jan | Feb | Mar | Apr | May | Jun | Jul | Aug | Sep | Oct | Nov | Dec |
|---|---|---|---|---|---|---|---|---|---|---|---|---|
| r_ALL | 0.17 | 0.14 | 0.36 | 0.31 | 0.25 | -0.16 | -0.56 | -0.37 | -0.75 | -0.48 | -0.19 | -0.03 |
| r_R | 0.29 | 0.15 | 0.38 | 0.40 | 0.23 | 0.49 | -0.15 | -0.07 | -0.67 | -0.46 | -0.18 | -0.03 |
| r_T | 0.11 | -0.06 | 0.02 | 0.09 | 0.30 | 0.05 | -0.64 | -0.30 | -0.48 | 0.08 | 0.08 | 0.05 |
| r_R_T_U | 0.10 | -0.05 | 0.00 | 0.11 | 0.33 | 0.35 | -0.20 | -0.08 | -0.13 | 0.10 | 0.09 | 0.06 |


## 550    4 Discussion

Our simulations suggested that annual ET and vegetation GPP were increasing over the

North China Plain during 1981 to 2013. Climate change contributed positive to ET
intensification, but it contributed negatively to GPP enhancement. Agronomical management
and technological advancement are the dominant factor to promote GPP increasing. The use
of remote sensing NDVI series have greatly improved the reliability of the vegetation water
consumption and productivity prediction at spatial and temporal scales, even if there were
uncertainty in vegetation characteristics retrievals from NDVI dataset. The results were
supported and consistent with most relevant studies at field and regional scales.

### 559    4.1 Is the trend of ET upward or downward over the NCP?

Although the crop productivities are steadily increasing, whether the actual ET over the

NCP is increasing or decreasing during the last three decades is controversial from the



reported literatures. By using the complementary relationship models (Brutsaert and Stricker,
1979), actual ET and potential ET both were decreasing (Cao et al., 2014; Gao et al., 2011).
However, ET was increasing predicted by the process - based VIP model from 1981 to 2013,
which was in consistent with the increasing trend of terrestrial greenness (Wang et al., 2016).
Yuan and Shen (2013) found that in the Northern part of NCP (Hebei Province) ET was
positively correlated with crop grain yield and agricultural water use was increasing from
2004 to 2008. Field measurements under well - watered fields also showed that seasonal ET
rates of both winter wheat and summer maize were increasing (Zhang et al., 2011). Bruatsaert
(2006) acknowledged that decreasing pan evaporation was an evidence of increasing
terrestrial evaporation. As general knowledge, the sign of ET change should be the same   as
that of vegetation greenness. Over the NCP a positive trend of ET was more believable, in the
light of significantly increasing NDVI over the growing season, especially in spring. The
positive effects of warming with higher water vapor deficit on ET might be offset by the
negative effect of declining solar radiation and wind speed on potential evaporation. On
viewpoint of the complementary relationship hypothesis (Hobbins et al., 2001), alteration of
available energy partitioned into latent heat flux (or ET) is dominated by the atmospheric
water vapor deficit. Namely, while more vapor is evaporated into the atmosphere boundary
layer, its water vapor deficit is correspondently relaxed, resulting in a lower rate of $ET_p$.
However, while declining global radiation being the dominant factor to ET trend, actual ET
($ET_a$), wet surface ET ($ET_w$) and potential ET ($ET_p$) are all tracing the trend of available
energy (net radiation). In the study period, net radiation was declining with a rate of -5.58 MJ
$yr^{-1}$ (r=0.56, p<0.01) over the NCP. As the trends of both $ET_p$ and $ET_w$ were dominated by the
radiation trend in the NCP, then $ET_a$ estimated from the complementary relationship was
definitely following the negative trend of radiation, because the positive trend of aerodynamic
evaporation was weak as a tradeoff of positive effect of rising water vapor deficit and
negative effect of decreasing wind speed. However, the declining ET trend resulted from the
reduced radiation has actually been reversed by increasing green leaf area, which would
reduce land surface albedo and temperature, etc. Consequently, ET and GPP were showing
slightly increasing. This study case also confirmed the limitations of the complementary
relationship for assessment of evaporation trend under the condition of radiaiton declining.





## 4.2 Effects of climate change and $CO_2$ on the cropping systems


Without adaptation measures, climate change is illustrated to exert negative effects on the
productivity of cropping system in the NCP during the last three decades (Mo et al., 2013; Liu et
al, 2010). Changes of individual climate variables affected differently on specific cropping
systems, associated with crop type and growing season. Climate warming in winter and spring
seasons was benefit to vegetative growth of winter wheat (Mo et al., 2013). Although air warming
has shortened the growing length, the autonomously adopted cultivars with higher thermal
requirement usually maintained the crop growth length and accumulated more photosynthesis
product, which may outweigh the extra respiration consumption under warmer climate (Wang et
al., 2010). So far as, the effects of global warming on wheat production were inhomogeneous in
the North China Plain, which were positive in the northern part but negative in the southern part
(Zhang et al., 2013). The reasons are that during the wheat growth period air temperature is still
below the favorable conditions in the high latitude part of the plain, thereby recent global warming
is benign to the wheat growth, however, the air warming may be detrimental in the southern part,
especially for rainfed wheat (Xiao and Tao, 2014).
However, dominated by summer monsoon in the North China Plain, climate is hot in summer
maize growth period. Due to maize is tropical originated species with high thermal requirement, it
can tolerate relative high air temperatures. Our study showed that it was not sufferred noticeably
from air warming in the recent decades, confirmed also by Xiong et al. (2012). However Guo et al.
(2014) reported that effect of air warming on maize was adverse with an Agro-Ecological Zones
model and the decreased daily temperature range (DTR) may be detrimental to crop yields.
Currently, adaptation measures may boost the production, such as harvest time delay (Wang et al.,
2014) or planting date advancement (Sacks and Kucharick, 2011) .
As increase of atmospheric aerosols by industrial production and combustion, global
radiation has declined in many parts of the world, in which direct component decreased but diffuse
component increased, so called "global dimming" (Liepert, 2002; Ren et al., 2013). The decline of
global radiation has resulted in less pan evaporation and carbon assimilation in crop and natural
vegetation communities (Xiong et al., 2012; Xiao and Tao, 2014), nevertheless plant canopies can
use the diffuse radiation with higher efficiency than direct beam (Gu et al., 2002). In spring time,
while the atmospheric circulation is shifting from continental to ocean monsoon in East Asia, the





wind speed is relative high and consiquently air pollution and aerosols are usually low, thereby
global radiation is not reduced obviously in the wheat growth period (Table 1), illustrating that air
warming and precipitation variability other than radiation decrease are the principal climate
factors contributing to the tendency of wheat production. In constrast, global radiation decline
significantly in summer season in the North China Plain (Table 1), as a result, productivities of
autumn harvest crops such as maize are mainly affected. For example, Guo et al. (2014) reported
that maize potential productivity was reduced by 20 kg hm$^{-2}$ due to global radiation decline in the
last decades over China.

During the study period of 1981 to 2013, atmospheric $CO_2$ concentrention increased from

340 to 396 ppm, which contributed to enhancement of crop productivity. However, most studies
with statistical analysis models neglected the contribution of $CO_2$ fertilization (e.g., Lobell and
Burke, 2010; Song et al., 2014). As confirmed by FACE experiments, elevated atmospheric $CO_2$
concentration is accelerating plant photosynthesis and reducing transpiration, whose fertilizer
effect is 0.065% per ppm increase for C3 plants (Field et al., 1995; Long et al., 2006; Ainsworth et
al., 2008). In our simulations, the contribution of $CO_2$ to GPP was positive while to ET was
negative. The positive $CO_2$ effect on GPP almost compensates for the negative effects of climatic
variable changes. However, we should bear in mind, as Ainsworth et al.(2008) pointed out that the
$CO_2$ fertilizer effect may be over-estimated by the process-based crop/ecosystem models.
**4.3 Effects of agronomical practice and technique advancement and other**
**factors**

Remotely sensed NDVI is an excellent indicator for long term changes of vegetation covers.

Here we assumed that climate change did not modify the tendencies of vegetation covers, but
dominated its inter-annual variation. Renewal of crop cultivars, applications of synthetic fertilizer
and irrigation, as well as conservancy tillage and nitrogen deposition are all contributing to the
crop and/or natural productivity improvements (Yu et al., 2012; Bai et al., 2015; Piao et al., 2015).
National statistical records of grain yields at county scale showed rapid increase from 1980 to
1990, and moderate increase in 2000s. Enhancement of crop yields was mainly stemmed from
more biomass accumulation and higher harvest index than previous varieties (Zhang X. et al.,
2013). In our simulations the upward trend of GPP was more significant than that of ET, which
was in consistent with the increasing trend of cumulative NDVI. Agronomical practices and



technology advancement contributed to 103% GPP changes in our study. By using crop models,
Yu et al. (2012) and Song et al. (2014) reported relative contributions of 92% and 62% by
agronomical management and renewal of cultivars for rice respectively, and Guo et al. (2014)
presented that 99.6 to 141.6% maize yield increases was contributed by technological
advancement in China since 1980s. The previous studies showed, if no adoption measures were
taken, climate change generally contributed negatively to crop productivities in the mid-latitude
areas, but the negative effects were usually compensated for by genetic improvements,
applications of fertilizer and irrigation, pest and weed control, as well as $CO_2$ and nitrogen
deposition effects (Liu et al., 2010; Lobell et al., 2011; Guo et al., 2014; Bai et al., 2015). Under
warming climate condition, it is expected that water requirement by crops and natural plants will
increase, but the intensified ET may be limited by insufficient soil moisture availability. Therefore,
the sustainability of crop productions are greatly depending on the improvement of agronomical
management and technological advancement on varietiy breeding.

## 5. Summary and conclusions

Climate change and human activities have greatly altered the hydrological regime and
crop productivity in the North China Plain with warm temperate climate during the recent
three decades. The VIP ecological model integrated the NOAA-AVHRR NDVI data series
predicted that spatial average annual actual ET was weakly increasing while vegetation
primary productivity (GPP) was significantly increasing ($p<0.01$) from 1981 to 2013, being
consistent with remotely sensed NDVI trend. The increases of actual ET and GPP were
mainly occurred in spring season, while ET and GPP were obviously decreasing in August
owing to global radiation diminishing.
Climate change, elevated atmospheric $CO_2$ fertilization and agronomical management all
contributed to the inter-annual trends of ET and crop GPP. The relative direct contributions of
climatic change, $CO_2$ fertilization and agronomical management to ET increasing were 56,
-28 and 68%, while the contributions to GPP were -32, 25 and 103%, respectively. Air
warming intensifies the crop water requirement and enhances the   production of crops
harvested in summer. The decline of global radiation resulted from exaggerated aerosol





concentration and air pollutions was considered to be the main cause of GPP reduction in
August. The study confirmed the necessary for imminent control of air pollution and aerosol
to sustain the agriculture system productivity.

### 683 Acknowledgements

This work was supported by the Natural Science Foundation of China grants (41471026
and 31171451). We thank the China Meteorological Administration (CMA) for providing the
meteorological data and on-site soil moisture data used in this paper.

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
