# Peer review of "Attributing Regional Trends of Evapotranspiration and Gross Primary Productivity with Remote Sensing: A case study in the North China Plain"

_Hydrology and Earth System Sciences, 2016_

## Referee Comment (RC1) · Anonymous Referee #1 · 13 Oct 2016

The authors conducted a study on the problem of climate and land use changes on vegetation ET using the VIP model. The manuscript is very interesting. But some problems should be improved regarding the robustness of the results. Major comments are given below.

Abstract L21: What does the word "management" mean?

Introduction L47-59 and L62~89: More thoroughly literature review is needed (such as Zhou et al. "Global pattern for the effect of climate and land cover on water yield." Nature communications 6 (2015); Chen et al. "50-year evapotranspiration declining and potential causations in subtropical Guangdong province, southern China." Catena 128 (2015): 185-194.). Authors should summarize the different change trends of ET

rather than listing some case studies. I also suggest that authors stated the pattern studies and mechanism studies separately. More potential influencing factors of ET changes should be mentioned.

Results analysis L272-281 Fig4: Even though the simulated GPP at the country scale showed good linear relationship with the statistic GPP. But, from Fig.4, we can clearly find the model overestimates 12%~130% of the GPP when GPP<1200, and underestimates 17%~58% of the GPP when GPP>2400 in the year 2000. The simulation error is also similar for the year 2005. The model might bring some uncertain influences on the trends of low and high GPP regions.

Fig.6, 7 and 10: I suggest the authors adding average NDVI, GPP, and ET of the study area.

---

## Referee Comment (RC2) · Anonymous Referee #2 · 25 Oct 2016

The manuscript analyzes the tendency of time-series GPP and ET simulated from VIP model, and the results agreed well with eddy covariance measurements. Quantitative analysis of the effects on the variations of GPP and ET are presented, which could provide scientific supports for the improvement of vegetation productivity and water use efficiency. The topic is relevant and suitable for HESS, however, the entire manuscript should be thoroughly revised and proofread (by native speakers).

MAJOR COMMENTS:

1. Multi-scale data were used in this study, for example, land cover classification (derived from Landsat TM and MODIS), NDVI products, meteorological data, eddy flux data, therefore, the most concerned issue is how authors dealt properly with the scaling

problem. More details of data preprocesses should be added in Section 2.3. 2. Different vegetation types have specific parameters in VIP model? Only farmland was considered in this research? The expressions and the specific parameters of VIP model used in this study should be provided. 3. Line 254-256: How did the authors calculate the biases of GPP and ET? The model predictions are affected by some associated uncertainties (input data, parameters, et al.). What are the effects of these uncertainties on the simulation results of GPP and ET? 4. The full names of abbreviated words should be provided for their first appearance, for example, VIP model. 5. All figures' types are not uniform, for example, font types are different; image scales are different. In fact, it would be better to draw the figures (plotting, bar charts) by some professional software (Origin, SigmaPlot, et al.).

SPECIFIC COMMENTS:

1. Line 191: "winther" or "winter". 2. Among the model outputs, one grid represents 8*8 km2, however the tower flux presents a small "footprint". How did the authors consider this issue? 3. Line 255: what is "absolute relative biases"? 4. Line 266: GPP performances were not shown in Fig.2. 5. Line 273: How was the yield data converted to GPP? Carbon content rates? 6. Line 395: Where is Fig.8b? 7. Line 405: please change the word "thant" to "than". 8. Line 408: The description of ET on spatial scale was shown in Fig.10. 9. Line 668-671: The sentence is confused. Please revise it. 10. The references format is confused.

Given these questions, I would recommend the manuscript with major revision.

Please also note the supplement to this comment:
http://www.hydrol-earth-syst-sci-discuss.net/hess-2016-419/hess-2016-419-RC2-supplement.pdf

---

## Author Comment (AC1) · 30 Nov 2016

Attributing Regional Trends of Evapotranspiration and Gross Primary Productivity with Remote Sensing: A case study in the North China Plain" by Xingguo Mo et al.

Reply to the referee one's comments:

Thanks very much for your valuable comments and suggestions. The reply to your comments is as the following:

Q: What does the word "management" mean?

A: Agricultural management practices include the schedules of irrigation and fertilization, replacement of new cultivars, etc. In our simulations, wheat and maize were irrigated according to rootzone water deficit. Nutrition deficit was not considered and assumed its effect are trivial.

Q: Introduction L47-59 and L62_89: More thoroughly literature review is needed.

A: We agree your suggestion. More thorough literature review will be carried out in the revision. The suggested two papers are very nice references for understanding the regional trend of evapotranspiration and its its influencing factors.

Q. Authors should summarize the different change trends of ET rather than listing some case studies. A: Thanks for your suggestion. Yes, we will summarize the ET trends with negative or positive signs in different region to clarify the determinant factor of long term water consumption.

Q: I also suggest that authors stated the pattern studies and mechanism studies separately. More potential influencing factors of ET changes should be mentioned.

A: The patterns and associated mechanism analysis are tightly related. The spatial and temporal variability of trends is obvious. The logic of result analysis is to present the spatial patterns of trends resulted from different factors, then try to figure out the contributing factors and their relative importance. We will make more clear the pattern and mechanism separation. The potential influencing factors of ET is complex. We will make more analysis and literature review on this aspect.

Q: Results analysis L272-281 Fig4: Even though the simulated GPP at the country scale showed good linear relationship with the statistic GPP. But, from Fig.4, we can clearly find the model overestimates 12%_130% of the GPP when GPP<1200, and underestimates 17%_58% of the GPP when GPP>2400 in the year 2000. The simulation error is also similar for the year 2005. The model might bring some uncertain influences on the trends of low and high GPP regions.

A: We noticed the prediction errors on the low and high yield counties are more significant, while comparing with the statistic crop yield at county scale. The yield at county level may involve some artifical biases in high and low productivity regions. In some high yield counties, their yields seems to be unreasonably higher than the nearby counties. We check the yield data in the revision. Also there may exist difference in crop productivity in high and low yield counties, which contributed to the prediction errors.

Q: Fig.6, 7 and 10: I suggest the authors adding average NDVI, GPP, and ET of the study area. A: Thanks, we will add the average NDVI, GPP and ET in the revision.

Reply to the referee two's comments:

Thanks very much for your constructive and valuable comments and suggestions. As we know, regional prediction of ecohydrological processes is still confronted with uncertainty from multiple aspects. VIP model used here is a physically based ecohydrological model, which has been validated and applied in the North China Plain and some other regions in a lot of researches. The modelled ET and GPP show great spatial variability. Land use in the plain is mainly consisted of cropland (more than 80%). The other land use/cover types are also modelled in the study. The separated contributions of evapotranspiration and gross primary production by climate change, elevated atmospheric $CO_2$ concentration and management seem reasonable, even if there are some uncertainties due to the model and remote sensing data deviations.

Q1. Multi-scale data wereused in this study,for example,land cover classification (derived from Landsat TM and MODIS), NDVI products, meteorological data, eddy flux data, therefore, the most concerned issue is how authors dealt properly with the scaling problem. More details of data preprocesses should be added in Section 2.3.

A1: Good comment. Land cover classification from the original Modis data product is 1 km resolution and Landsat TM landuse data is 1:100, 000. TM data is resampled to 1km with the majority method. We use rice paddy class in TM landuse data to replace the corresponding farmland pixel in Modis land cover data, so that the farmland is set as rice paddy and non-rice farmland. Then the farmland is resampled to 8km. We also

calculate the fractions of land cover types in a pixel. NDVI products of AVHRR are 8km resolution, which smoothed by S-G filter to remove the unreasonable low values but keep the high values.

Daily records of meteorological stations in and around the study area are interpolated to 8km grids with the gradient inverse distance square method (GIDS) which takes into account the multiple regression between the climate variables and elevation, lattitude and longitude.

For model prediction validated with the eddy covariance flux measurements, we drive the model with the local meteorological mask measurements. Thence, the scale inconsistency is avoided while comparing the model predictions and the measurements.

Q2. Different vegetation typeshave specificparameters in VIP model? Only farmland was considered in this research? The expressions and the specific parametersof VIP modelused in this studyshould be provided.

A2: Yes, vegetation types have their specific parameters in the model. All the types are considered, and irrigation farmland and rainfed farmland are also identifed. As you suggestion, the specific parameter of VIP model will be listed in table in the revised manuscript.

Q3. Line 254-256: How did the authors calculate the biases of GPP and ET? The model predictions are affected by some associated uncertainties(input data, parameters, et al.). What are the effects of these uncertainties on the simulation results of GPP and ET?

A3: We calculated the annual biases of model predicted and measured GPP and ET at annual scale, in addition to the daily scale. Maybe the original sentence is not expressed clear and make you confused. The sensitivity of model prediction to the parameters and input meteorological data are discussed in another paper (Mo et al., EMS; Mo et al., IJClimatol). As shown in our previous study, the effects of parameter

uncertainties in their possible range may be as high as 16.5% and 21.1% for ET and GPP respectively. But while the key parameters are tuned according to the measurement, the model accuracy will significant improved.

Q4: The full namesof abbreviated words should be provided for their first appearance,for example, VIPmodel.

A4: Thanks. We will do it in the revision.

Q5. All figures' types are not uniform, for example, font types are different; image scales are different. In fact, it would be better to draw the figures (plotting, bar charts) by some professional software (Origin, SigmaPlot, et al.).

A5: Thanks. We will redraw all the figures and maps.

SPECIFIC COMMENTS:

Q1. Line 191: "winther"or"winter".

A1: Thanks.

Q2. Among the model outputs, one grid represents 8*8 km2, howeverthe tower flux presentsa small "footprint". How did the authors consider this issue?

A2. Yes, it is a hard question. We did not compare directly the 8km grid flux with tower flux. But while the landscape is flat and land cover is relative homogeneous, the 8km grid flux is shown to have high correlation with tower fluxes. We did that kind of comparison in another study.

Q3. Line 255: what is "absolute relative biases"?

A3: It should be expressed as relative absolute bias or relativee absolute error.

Q4. Line 266: GPP performances were not shown in Fig.2.

A4: It is shown in Fig.3

Q5. Line 273: How was the yield data converted to GPP? Carbon content rates?

A5: Yield was converted to GPP according to the carbon content, harvest index, root to shoot ratio, NPP/GPP ratio, moisture in the grain, etc.

Q6. Line 395: Where is Fig.8b?

A6: Sorry, Fig8a and Fig8b should be Fig9a and Fig9b respectively.

Q7. Line 405: please changethe word "thant"to "than".

A7: Thanks.

Q8. Line 408: The description of ET on spatial scale was shown in Fig.10.

A: We will redraw this image.

Q9. Line 668-671: The sentence is confused. Please revise it.

A9. We will rewrite this sentence.

Q10. The references format is confused.

A10. We prepare the reference format according to a recent HESS paper. We will check the reference format again.